# Coir, an Alternative to Peat—Effects on Plant Growth, Phytochemical Accumulation, and Antioxidant Power of Spinach



**Rui M. A. Machado [1,\*], Isabel Alves-Pereira [2] , Rui Ferreira [2,\*] and Nazim S. Gruda [3]**

[1] MED—Mediterranean Institute for Agriculture, Environment and Development, Departamento de Fitotecnia, Escola de Ciências e Tecnologia, Universidade de Évora, 7006-554 Évora, Portugal

[2] MED—Mediterranean Institute for Agriculture, Environment and Development, Departamento de Química, Escola de Ciências e Tecnologia, Universidade de Évora, 7000-671 Évora, Portugal; iap@uevora.pt

[3] Division of Horticultural Sciences, University of Bonn, Auf dem Hügel 6, 53121 Bon, Germany; ngruda@uni-bonn.de

\* Correspondence: rmam@uevora.pt (R.M.A.M.); raf@uevora.pt (R.F.)

**Abstract:** The effects of four commercial substrates, a peat-based substrate, and three coir types (coir pith, coir chips, and coir pith + fibers) on yield, phytochemical accumulation, and antioxidant activity were evaluated in *Spinacia oleracea* L. cv. 'Manatee'. Soil-blocked spinach seedlings were transplanted into Styrofoam planting boxes filled with the substrate. Each planting box was irrigated daily by drip with a complete nutrient solution, and the irrigation scheduling was optimized to the peat. Leaf area and fresh yield in coir pith and coir pith + fiber were similar to those obtained in peat. However, shoot dry weight accumulation and leaf chlorophyll were lower in plants grown in coir. Substrate type did not affect leaf carotenoids. Total flavonoid content was higher in plants grown in the different types of coir. Total phenols and antioxidant activity (DPPH) were higher in plants grown in coir pith. This indicates that the different coir types, mainly coir pith, may provide an alternative to peat since they allowed a high fresh yield to be reached and the total flavonoids to be increased. In contrast, the levels of other phytochemicals and antioxidant activity were usual for spinach. However, further research is necessary to analyze the effects of irrigation scheduling and the nutrient solution adjusted to each growing medium on yield and phytochemical accumulation.

**Keywords:** *Spinacia oleracea*; substrates; soilless culture systems; photosynthetic pigments; phenols; flavonoids; ascorbic acid; DPPH; FRAP

## 1. Introduction

Peat alone or mixed with other constituents is the most used material in horticultural production. However, peat is a nonrenewable resource. Its exploration has negative environmental and ecological impacts [1,2], being classified as the growing medium with the greatest impact on climate change and resources [3]. Coir, also known as coir dust, coir meal, coir pith, and coir fibers, may provide an alternative to peat since it is a biodegradable and renewable by-product. Social and ecological questions concerning child labor, inadequate wastewater management, and transportation should be additionally considered. From the perspective of substrate properties, coir pith has high water capacity and easily available water. It contains more lignin and less cellulose than peat, thus being more resistant to microbial breakdown. It is also easily rewettable, which improves the water absorption of substrate mixtures and water distribution in the growing medium [2,4]. All those properties make coir pith a good peat alternative growing medium. The use of coir enabled high yields in spinach [5,6]. However, nowadays, in addition to yield, the nutritional quality of vegetables is essential. A further increase in bioactive compounds is desirable and an object of diverse research projects worldwide.

Spinach is one of the healthiest vegetables for the human diet due to its high concentration of nutrients and health-promoting compounds [7]. Among vegetable crops, it has one of the highest aggregate nutrient density index values [8,9].

According to [10], the nutrient level in plants is strongly affected by nutrient solution characteristics, such as the nutrient concentration, chemical forms of the elements, the temperature of the nutrient solution, pH, and irrigation scheduling (dose and frequency). On the other hand, substrate characteristics and irrigation interaction influence wetting and salt patterns in the root medium, easily available water content, leaching fraction, and nutrient and water availability. This affects the water and nutrient uptake by plants, which may lead to a greater or lesser degree of abiotic stress related to the water deficit, nutrient deficiency, salinity, and the combination of these factors.

Abiotic stress affects phytochemical accumulation and antioxidant activity. In response to water deficit, plants typically accumulate phytochemicals of low molecular weight and enzymes for scavenging the reactive oxygen species (ROS) induced by stress. [11,12].

The synthesis and accumulation of secondary metabolites may be associated with changes in nutritional status [13,14]. Thus, nitrogen, potassium, and phosphorus deficiency affect phytochemical accumulation in spinach [15]. Salinity affects the bioactivity of various fruits and vegetables and could be considered a sustainable and low-cost approach towards this direction [14]. Cultural practices that involve either low fertilizer levels or slight and moderate salt stress could reduce the yield but improve the nutritional value of vegetables [10,16], including spinach [15]. According to Shimomachi et al. [17], salt stress increased polyphenol contents in spinach. However, [18] reported that moderate levels of nutrient solution concentration (1.2 and 1.7 dS m$^{-1}$) did not affect total phenols, ascorbic acid, chlorophyll a and b, carotenoids, and ascorbate peroxidase content. It could be concluded that the response of phytochemical accumulation to salinity is not always linear [19] and clear [20].

The physicochemical properties of coir in the market differ significantly from peat [21]. This is due to different levels of fiber, which may affect water and plant nutrition, creating a greater or lesser abiotic degree of stress. Therefore, we hypothesize that coir can replace peat, but it is necessary to know their effects on yield and nutritional quality of the produce.

Therefore, this study aimed to evaluate the effects of different coir types on plant growth and nutritional quality, such as phytochemical composition, antioxidant enzyme levels, and antioxidant activity of spinach grown during late winter and early spring in unheated greenhouses.

## 2. Materials and Methods

### 2.1. Growth Conditions and Substrates

The experiment was conducted in a greenhouse located at the "Herdade Experimental da Mitra" (38°31′52″ N; 8°01′05″ W), University of Évora, Portugal. The greenhouse was covered with polycarbonate and had no supplemental lighting or heating. Diurnal changes in air temperature inside the greenhouse at the plant canopy level ranged from 8 to 27 °C. Solar radiation ranged from 34 to 248 W·m$^{-2}$·d$^{-1}$.

Our experiment used four commercial substrates: peat (70% black peat + 30% white peat) and three different types of coir from Projar Group (Table 1). According to the manufacturer, coir chips, coir pith, and coir pith + fiber had 0, 7, and 20% fiber, respectively.

Spinach (*Spinacia oleracea* L. cv. Manatee) seedlings were produced in soil blocks with six seedlings per block 18 days after emergence. Soil blocks were obtained from a commercial nursery. They were transplanted into Styrofoam plant boxes on 16 February 2017. The boxes (100 × 25 × 10 cm) were filled with 14 L substrate at the height of approx. 7 cm. The blocks were spaced 12.5 cm in two rows per box and 10 cm between rows with a plant density of 384 plants m$^{-2}$. Treatments were arranged in a complete randomized block design with five replicates. Each planting box was irrigated using 4 L·h$^{-1}$ pressure-compensating and antidrain emitters. The emitters were attached to 4 fine tubes with 70 cm

length and 5 mm diameter, inserted into the substrate along the center of the Styrofoam box. Thus, 8 water emission points were used per box.

**Table 1.** Physicochemical properties of substrates.

| Substrate | Peat | Coir Pith | Coir Chips | Coir Pith + Fiber |
|---|---|---|---|---|
| Composition | 70% black peat + 30% white peat | 100% | 100% | 93% coir pith + 7% fiber |
| pH * | 5.5–6.0 | 5.5–6.0 | 5.5–6.2 | 5.5–6.2 |
| EC (dS·m$^{-1}$) * | 1.5–1.8 | <1.9 | ≥1.5 | ≥1.5 |
| CEC (meq/100g)* | 100–190 | 60–120 | 20–40 | 40–80 |
| N (mg L$^{-1}$) * | 50–300 | | | |
| P (mg L$^{-1}$) * | 35–131 | | | |
| K (mg L$^{-1}$) * | 60–330 | | | |
| Total porosity (*v/v*, %) * | | 95 | | |
| Granulometry (mm) * | | 0–10 | 10–15 | 2–4 |
| Air (*v/v*, %) * | - | 25 | 40 | 30 |
| Water holding capacity (*v/v*, %) * | - | 70 | 54 | 65 |
| Mass wetness (g water/g substrate) ** | 6.07 $^{\dagger}$ c | 7.84 b | 5.75 d | 8.65 a |
| Moisture content (*w/w*, %) ** | 82.6 ab | 84.68 a | 71.10 b | 84.63 a |
| Bulk density (g·cm$^{-3}$) ** | 0.127 a | 0.103 a | 0.070 b | 0.081 a |

* According to the manufacturer. ** Determined following the methods described in [22]. Moisture content: The percent moisture found in a sample on a wet mass basis. This is calculated by ((wet weight − dry weight)/wet weight) × 100. Mass wetness the water content of a sample on a dry mass basis. This is calculated by (wet weight − dry weight)/dry weight. $^{\dagger}$ Means followed by different letters within a line are significantly different at $p < 0.05$.

The irrigation schedule was optimized for peat. It was based on substrate volumetric water content at Styrofoam box control (peat), measured using a soil moisture probe (SM105T delta devices England), and the volume of water drained.

The nutrient solution was applied three to seven times per day, depending on climatic conditions, and averaged 15 to 30% drainage, i.e., leaching fraction, for each application. The leaching fraction was controlled through a relay level connected to an electric valve that stopped watering when the level of leached water was within 10 to 25% of the applied water. Excepting the first irrigation to moisten the growing mediums, the nutrient solution was applied continuously from transplanting to the day before harvesting.

The fresh tap water had an electrical conductivity (EC) of 0.4–0.5 dS·m$^{-1}$ and a pH of 7–7.4 and contained 0.10–0.30 mol·L$^{-1}$ NO$_3$, 1 mol·L$^{-1}$ Ca, 1 mol·L$^{-1}$ Mg, 2.1 mmol·L$^{-1}$ Cl$^{-}$, 0.7 mol·L$^{-1}$ Na, 0.53 μmol·L$^{-1}$ Fe, and 0.16 μmol·L$^{-1}$ Mn. The nutrient solution initially contained 7.21 mol·L$^{-1}$ NO$_3$, 2.32 mmol·L$^{-1}$ NH$_4$, 0.59 mmol·L$^{-1}$ P, 3.38 mmol·L$^{-1}$ K, 2.55 mmol·L$^{-1}$ Ca, 1.35 mmol·L$^{-1}$ Mg, 0.80 mmol·L$^{-1}$ S, 46 μmol·L$^{-1}$ B, 7.86 μmol·L$^{-1}$ Cu, 8.95 μmol·L$^{-1}$ Fe, 18.3 μmol·L$^{-1}$ Mn, 1 μmol·L$^{-1}$ Mo, 2 μmol·L$^{-1}$ Zn, 2.1 mmol·L$^{-1}$ Cl$^{-}$, and 0.7 mmol·L$^{-1}$ Na.

At 26 DAT, in order to reduce the nitrate concentration in the leaves, the nutrient concentrations and the NO$_3$/NH$_4$ ratio in the nutrient solution were adjusted to 4.26 mmol·L$^{-1}$ NO$_3$, 4.11 mmol·L$^{-1}$ NH$^4$, 0.67 mmol·L$^{-1}$ P, 2.84 mmol·L$^{-1}$ K, 2.13 mmol·L$^{-1}$ Ca, 0.88 mmol·L$^{-1}$ Mg, 0.47 mmol·L$^{-1}$ S, 46 μmol·L$^{-1}$ B, 7.86 μmol·L$^{-1}$ Cu, 8.95 μmol·L$^{-1}$ Fe, 18.3 μmol·L$^{-1}$ Mn, 1 μmol·L$^{-1}$ Mo, 2 μmol·L$^{-1}$ Zn, 2.1 μmol·L$^{-1}$ Cl$^{-}$, and 0.7 mmol·L$^{-1}$ Na.

*2.2. Measurements*

The pH, EC, and the concentration of NO$_3$ of the drainage water from each box were measured weekly using a potentiometer (pH Micro 2000 Crison), a conductivity meter (LF 330 WTW, Weilheim, Germany), and an ion-specific electrode (Crison Instruments, Barcelona, Spain), respectively, following the procedures outlined in [23].

The plants were harvested at 40 DAT. The shoots of the plants were cut off at 1 cm above the substrate surface. The shoots of five representative plants from each box were washed, oven-dried at 70 °C for 2–3 days, weighed, and ground.

Samples of 1.000 g of spinach leaf-blade from four treatments and five replicates were macerated in a mortar and homogenized in 8 mL of methanol/water solution (90:10 ($v/v$), MW90 extract) for 1 min and then centrifuged at 4 °C at 6440× $g$ for 5 min. The methanol extracts were stored in aliquots at −20 °C for later use [24]. Total chlorophyll, chlorophyll a (Chl a) and b (Chl b), and total carotenoids (Cc) were determined in MW90 extract by the method of [24] using the following equations:

$$\text{Chl a } (\mu g/mL) = 16.82 \text{ A665.2} - 9.28 \text{ A652.4};$$

$$\text{Chl b } (\mu g/mL) = 36.92 \text{ A652.4} - 16.54 \text{ A665.2};$$

$$\text{Cc } (\mu g/mL) = (1000 \text{ A470} - 1.91\text{Chl a} - 95.15\text{Chl b})/225,$$

where A = absorbance, Chl a = chlorophyll a, Chl b = chlorophyll b, and Cc = carotenoids.

Samples of 1.000 g of spinach leaf-blade were macerated in a mortar and homogenized in 8 mL of methanol/water solution (80:20 ($v/v$), MW80 extract) for 1 min and then centrifuged at 4 °C at 6440× $g$ for 5 min. The methanol extracts were stored in aliquots at −20 °C for later use.

Content of total phenolic compounds (TPCs) was determined using Folin–Ciocalteau phenol reagent described earlier [25], reading the absorbance at 760 nm. TPC content expressed as milligrams of gallic acid equivalent (GAE) per 100 g of fresh weight (FW) was calculated using a calibration curve (GAE, $n = 6$ concentrations from 0 to 50 mg/L).

For determination of flavonoid contents, 100 μL of MW80 extract was mixed with 20 μL of 10% $AlCl_3$ ($w/v$), 500 μL of 1 M potassium acetate, and 380 μL of distilled water and incubated at 25 °C for 30 min. Total flavonoid content was determined by reading the absorbance at 420 nm, using an extinction coefficient of 0.004 $\mu M^{-1}$ $cm^{-1}$, and expressed in mg of quercetin equivalent (QE) per 100 g of fresh weight [26].

Total anthocyanin content was determined by mixing 500 μL of MW80 extract with 500 μL of 50% ethanol ($v/v$) and 84 μL of 37% HCl. After incubation at 60 °C for 30 min, the absorbance was measured at 530, 620, and 650 nm, and the absorbance of cyanidin-3-glycoside was calculated using the following equation:

$$\text{Aant} = (\text{A530} - \text{A620}) - 0.1 \,(\text{A650} - \text{A620}).$$

Total anthocyanin content was calculated using a molar extinction coefficient of 34,300 $M^{-1}cm^{-1}$ and a molecular weight of 449.2 $gmol^{-1}$ and expressed in mg of cyanidin-3-glycoside equivalent (C3GE) per 100 g of fresh weight [27].

Ascorbic acid (AsA) content was determined by the method of [28], incubating the sample (extracts or standard suitably diluted) in a mixture containing 5% TCA in ethanol, 0.4% $H_3PO_4$, 0.5% β-phenanthroline in ethanol, and 0.03% $FeCl_3$ in ethanol, warmed at 30 °C, for 90 min. The absorbance of Fe (II)–β-phenanthroline complex formed was read at 534 nm. AsA concentration was calculated using a calibration curve (ascorbic acid, $n = 6$ concentrations from 0 to 30 mg/L).

Free Pro levels of MW80 extract were determined using the acid ninhydrin reaction [29], reading the absorbance of yellow-orange chromophore formed 546 nm. Pro concentration was calculated using a calibration curve (L-proline, $n = 6$ concentrations between 0 and 20 mg/L).

The 2,2-diphenyl-1-picrylhydrazyl free radical scavenging antioxidant power (DPPH) was determined by measuring the ability of plant MW80 extracts to capture the stable organic radical DPPH$^\bullet$ (2, 2-diphenyl-1-picryl-hydrazyl, violet) and its conversion into a stable product, DPPH-H (diphenyl-picryl hydrazine, yellow). Aliquots of an extemporaneous methanol solution of 0.03 g/L DPPH$^\bullet$, kept in the dark, were added to a known volume of sample or standard solution. The reduction of DPPH$^\bullet$ to DPPH-H was followed

by reading the absorbance at 515 nm, at 25 °C, for 180 s. Antioxidant power reported as milligrams of GAE per 100 g of FW was calculated using a calibration curve (GAE, $n = 8$ concentrations from 0 to 200 mg L$^{-1}$) [30].

Ferric reducing antioxidant power (FRAP) was determined by the method of [25]. In sum, the FRAP reagent was prepared freshly by mixing 300 mM acetate buffer pH 3.6 and 10 mM TPTZ solution in 40 mM HCl and 20 mM iron (III) chloride solution (10:1:1, $v/v/v$) and warmed to 37 °C before use. Then, 0.050 mL of the sample (suitably diluted MW80 extracts or standard) was mixed with 0.950 mL of FRAP reagent. Absorbance change was read at 593 nm at 37 °C, for 180 s. The reducing power of iron present in the samples reported as milligrams of Trolox equivalent per 100 g of FW was calculated using a calibration curve (Trolox solution, $n = 8$ concentrations from 0 to 1120 mg L$^{-1}$). For all previous determinations, a Genesys10S UV/Vis spectrophotometer was used.

Samples of 1.000 g of spinach leaf blade were macerated in liquid $N_2$ and homogenized in 5 mL of 0.12 mM phosphate buffer pH 7.2. The obtained supernatant using the centrifugation of this extract for 15 min at 15,000× $g$ at 4 °C was collected and stored in aliquots at −20 °C (PB extract) for further use [31].

Glutathione (GSH) was assayed by the method of [32], based on the reaction of o-phthalaldehyde (OPT) as a fluorescent reagent with GSH at pH 8 present in the PB extract. The fluorescence of products was determined at 420 nm with the excitation at 350 nm, at 25 °C, using GSH as a standard in a single-beam Shimadzu RF-5001PC fluorimeter.

Glutathione reductase (GR) enzyme activity was determined by the method of [33] in a reaction mixture containing 15 mM EDTA, 635 mM GSSG, and a suitable volume of leaf-blade PB extract (0.5–0.2 mg mL$^{-1}$ protein) in 0.12 mM phosphate buffer pH 7.2. The reaction was started with the addition of 9.6 mM NADPH. The oxidation of NADPH was determined by reading the absorbance at 340 nm for 360 s. At 37 °C, GR activity was calculated based on the slope of the reaction curves, using an extinction coefficient value of 6.22 mM$^{-1}$ cm$^{-1}$ for NADPH. GR activity was expressed in terms of nmol min$^{-1}$/mg protein.

Peroxidase enzyme activity (POD) was determined by the method of [34] in a reaction mixture containing 1% p-phenylenediamine, 1.5% hydrogen peroxide, and a suitable volume of leaf-blade PB extract (0.5 mg/mL of protein) in 0.2 M potassium phosphate buffer pH 6.5. The oxidation of p-phenylenediamine was determined by reading the absorbance at 485 nm for 10 min, at 25 °C. POD activity was calculated based on the slope of the reaction curves using the value of the extinction coefficient of $2.1 \times 10^4$ M$^{-1}$ cm$^{-1}$ for p-phenylenediamine. For all enzyme determinations, a double-beam Hitachi-U2001 spectrophotometer with temperature control was used.

The protein content of the PB extract was determined by the method of [35], using a calibration curve (bovine serum albumin (BSA); $n = 6$ concentrations from 0 to 200 μg mL$^{-1}$).

Data were analyzed using the analysis of variance using SPSS Statistics 25 software (Chicago, IL, USA). Means were separated at the 5% level using Duncan's new multiple range test. Bivariate correlation analysis between parameters was realized using Pearson's bilateral correlation coefficient.

## 3. Results and Discussion

### 3.1. Drainage Water

Nitrate and $H_3O^+$ concentration in the drainage water were affected by the substrate (Figure 1). Nitrate concentration in the drainage water was higher in peat until 15 DAP and in coir-chips during the crop cycle than in the other treatments (Figure 1). Coir chips had the highest drainage volume (data not presented) and the lowest wetness mass (5.75 g water/g substrate). In general, the $H_3O^+$ concentration was lower in peat and coir chips than in the other coir types. The differences could be due to the different cation exchange capacities of the substrates that contribute to the adsorption of the hydronium ions. The differences in nitrate leaching can also affect the form of nitrogen uptake by plants, affecting

hydronium and hydroxide concentration in the root medium. The EC of the drainage water was not significantly affected by substrate type.

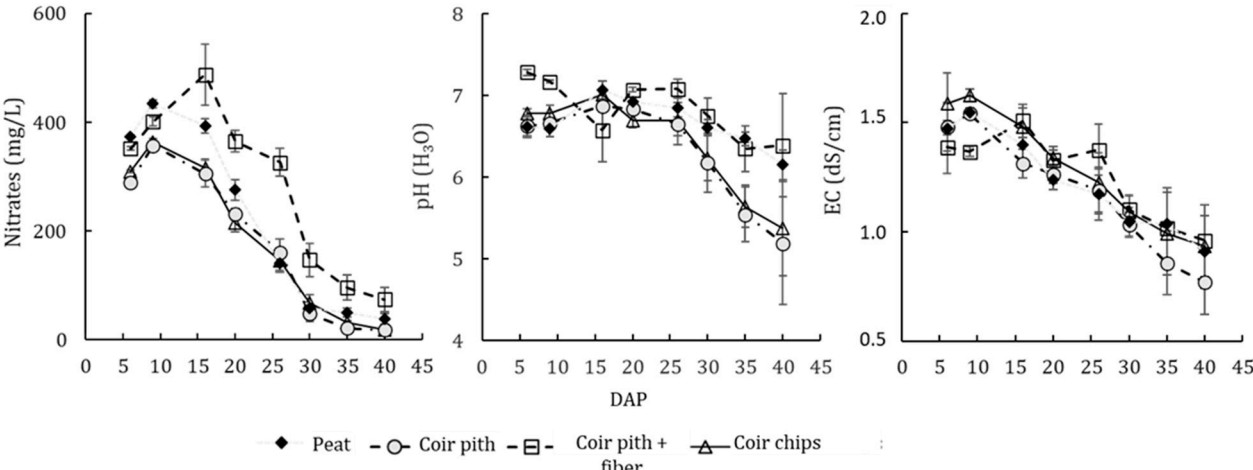

**Figure 1.** The $NO_3$ and $H_3O^+$ concentrations and EC in the drainage water. Each symbol represents the mean of five replicates, and the error bars represent ± 1SE.

$NO_3^-$ concentration and the EC in the drainage water on the last three sampling dates, as compared with the $NO_3^-$ and the EC in the nutrient solution, decreased significantly (Figure 1) due to the decrease in nitrate applied to the nutrient solution (264 mg $NO_3^-$ $L^{-1}$) and due to high nutrient uptake by spinach plants.

### 3.2. Plant Growth and Yield

Leaf area and spinach fresh yield in coir pith and coir pith + fiber did not differ significantly from those obtained in peat (Figure 2). The coir pith and coir-pith + fiber yields were high, ranging from 3.79 to 4.32 kg·m$^2$. These findings are consistent with those obtained in [5,6]. The use of coir, a growing medium, enables the achievement of high yields. The fresh yield in coir chips (2.64 kg m$^{-2}$) was lower than in the other substrates.

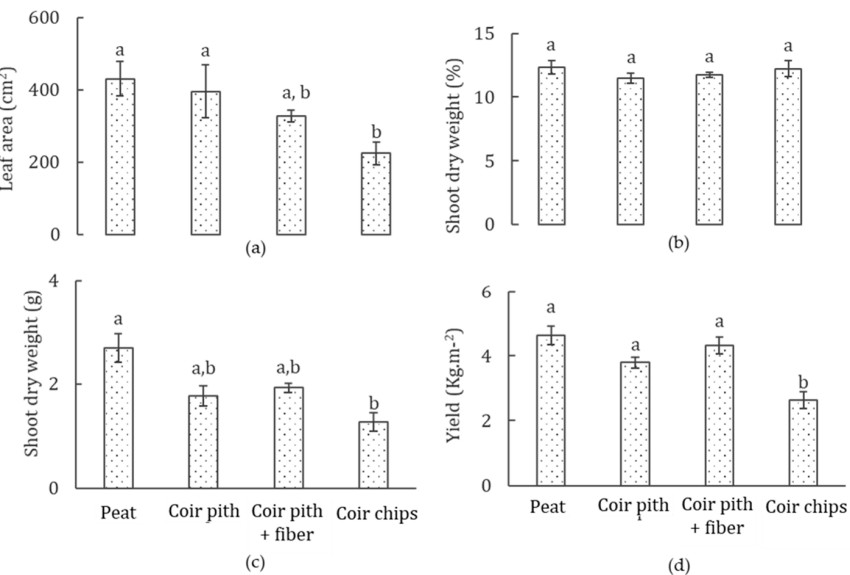

**Figure 2.** Leaf area (**a**), shoot dry weight (%) (**b**), shoot dry weight per plant (**c**), and fresh yield (**d**) of spinach. Means with different letters are significantly different at $p < 0.05$. Each bar represents the mean of five replicates, and the error bars represent ± 1SE.

Plants grown in coir accumulated less shoot biomass than those grown in peat. This could be due to the physiochemical properties of coir, with lower nutrient and water holding capacity than peat. Moreover, the initial content of nutrients in peat (Table 1) and the substrates' interaction and irrigation frequency affect substrate water content and nutrient availability [36]. Shoot biomass accumulation was positively correlated to leaf K (r = 0.75, $p > 0.01$) and N (r = 0.58 $p > 0.01$) content, with a higher level in plants grown in peat than in plants grown in coir (Table 2). Biomass accumulation and crop growth are related to crop N accumulation [37]. In spinach, shoot biomass decreased in response to deficit irrigation [38], and it was higher in the plants grown in coir, mainly in coir chips.

### 3.3. Leaf Nutrients

Leaf nutrient concentrations of the plants grown in coir pith, coir pith + fiber, and peat did not differ significantly, except for the concentrations of potassium and calcium. Leaf N, K, Mn, and Zn concentrations in plants grown in coir chips were lower than those grown in the other substrates (Table 2). However, leaf Ca and Mg concentrations were higher in plants grown in coir chips than in plants grown in the other substrates. That could be related to the low bulk density of the substrate (Table 1), which may have allowed for high root branching and Ca and Mg uptake primarily occurring in the new roots [39].

Despite some differences, the concentrations of macronutrients, except nitrogen, were within the sufficiency ranges (Table 2). Leaf nitrogen average values in coir pith + fiber and coir chips were slightly lower than the lower end of the sufficiency range (4%). However, the plant shoot dry weight of the plants grown in peat was higher, and plant nutrient uptake may also have increased. Shoot nutrient uptake in spinach increased with dry shoot matter in plants with the same leaf nutrient content [14].

Leaf Fe, B, Cu, and Mn concentrations were unaffected by substrate type. Leaf Zn content was well above the recommended range (Table 2), which could be due to the $NH_4^+$ concentration in the nutrient solution, which was high from 26 DAT until the harvest. In lettuce, Savvas et al. (2006) [40] reported an increase in leaf Zn as ammonium supply increased. These Zn concentrations are higher than the sufficiency range (100 μg g$^{-1}$ DM [41] and 75 μg g$^{-1}$ DM [42]). However, none of the plants in the treatments showed visual symptoms of excess Zn. Zinc in excess can cause chlorosis in leaves due to a reduction in chlorophyll [43]. According to [44], leaf Zn concentrations of up to 100–700 mg kg$^{-1}$ DM can be achieved without yield loss, which can be advantageous since Zn is a desirable nutrient for human health.

**Table 2.** Nutrient concentration in fully expanded spinach leaves.

| Substrate | Leaf Macronutrients (%) | | | | | Leaf Micronutrients (μg·g$^{-1}$) | | | | |
|---|---|---|---|---|---|---|---|---|---|---|
| | **N** | **P** | **K** | **Ca** | **Mg** | **Fe** | **B** | **Cu** | **Mn** | **Zn** |
| Peat | 4.48 a [Z] | 0.38 | 6.88 a | 0.94 c | 0.70 b | 89.2 | 34.0 | 31.2 | 105.2 a | 198.2 a |
| Coir pith | 4.18 a | 0.36 | 5.96 b | 1.00 b | 0.64 b | 112.6 | 33.2 | 39.4 | 104.8 a | 211.0 a |
| Coir pith + fiber | 3.98 ab | 0.32 | 6.26 ab | 1.04 ab | 0.72 b | 77.0 | 35.2 | 25.4 | 104.8 a | 210.0 a |
| Coir chips | 3.48 b | 0.32 | 5.14 bc | 1.20 a | 0.82 a | 65.0 | 30.0 | 29.6 | 73.0 b | 115.2 b |
| Recommended range | | | | | | | | | | |
| [41] | 4.00–6.00 | 0.30–0.60 | 5.00–8.00 | 0.70–1.20 | 0.60–1.00 | 60–200 | 25–60 | 5–25 | 30–250 | 25–100 |
| [42] | 4.00–6.00 | 0.30–0.50 | 3.00–8.00 | 1.00–1.50 | 0.40–1.00 | 50–200 | 25–60 | 5–15 | 25–200 | 20–75 |

[Z] Means followed by different letters within a column are significantly different at $p \leq 0.05$.

### 3.4. Photosynthetic Pigments

Leaf total chlorophyll (chl a + chl b) (Figure 3a) and chlorophyll b (Figure 3c) were higher in plants grown in peat and coir chips than in plants grown in coir pith and coir pith + fiber. Total chlorophyll in a plant grown in peat (79.82 mg/100 g FW) was similar to that recorded in [45] (65.4 mg/100 g FW) and [46]. However, it was lower than those reported in [47] (96.2 to 301.8 mg/100 g FW). Conversely, the ratio of chlorophyll a to chlorophyll b (Figure 3d) was higher in plants grown in coir pith and coir pith + fiber. The differences in

chlorophyll could be due to the water availability; salinity in the root media; and nutrient uptake for nitrogen, potassium, and zinc. These factors or their combination may affect chlorophyll biosynthesis.

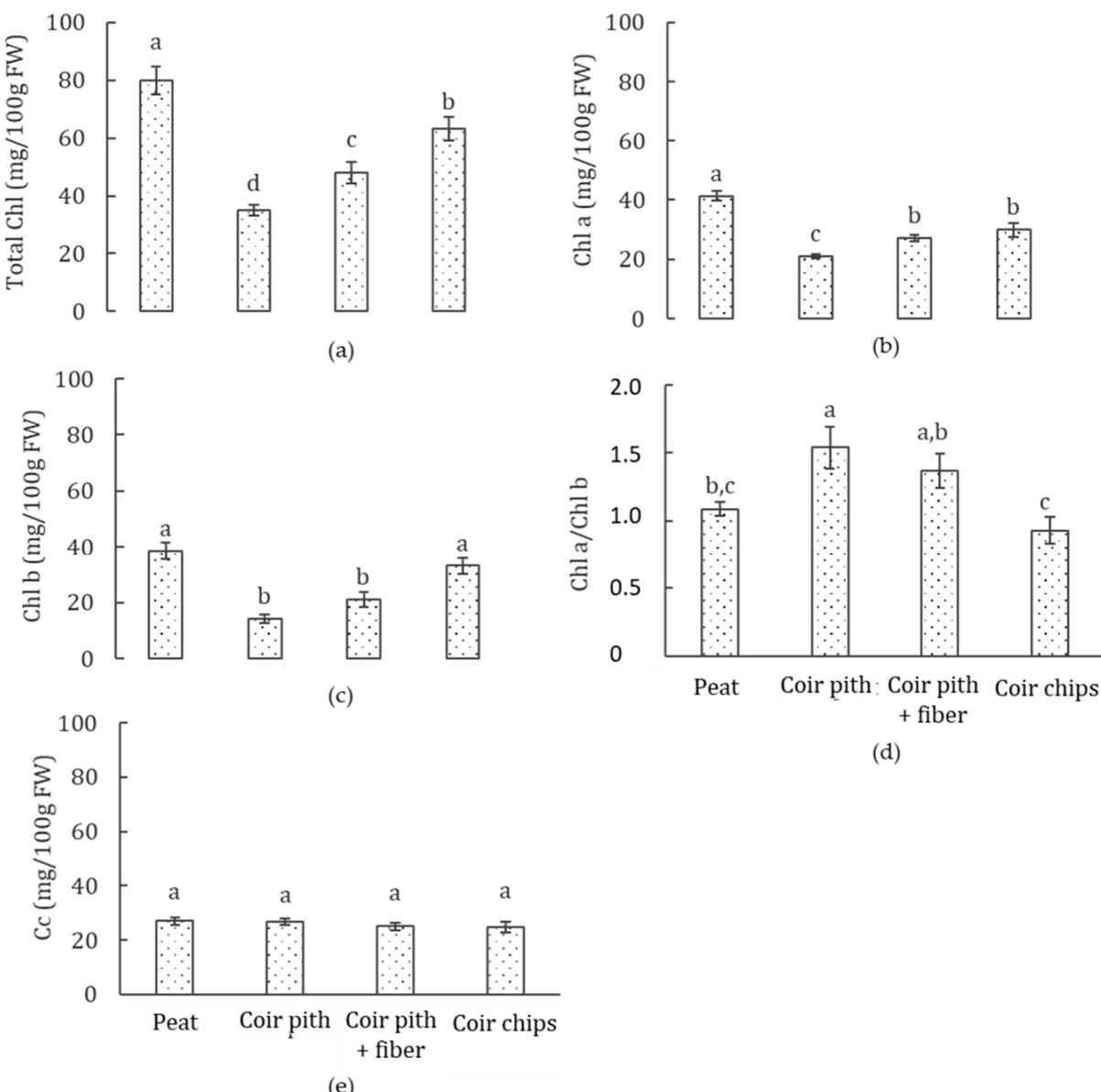

**Figure 3.** Accumulation of photosynthetic pigments (total chlorophyll (**a**), Chl a (**b**), Chl b (**c**), and carotenoids (Cc) (**e**)) and Chl a/Chl b ratio (**d**). Means with different letters are significantly different at $p < 0.05$; FW—fresh weight. Each bar represents the mean of five replicates, and the error bars represent ± 1SE.

Abiotic stresses have negative influences on chlorophyll biosynthesis [48]. Salinity reduces the contents of photosynthetic pigments [49,50]. Average levels of chlorophyll were lower in plants with average values of Zn > 200 μg g$^{-1}$, that is, in plants grown in coir pith and coir pith + fiber. As previously mentioned, high levels of Zn in spinach can decrease the chlorophyll content.

Leaf-blade Cc content was unaffected by substrate type (Figure 3e). According to [51], leaf Cc of strawberries was not also affected by substrate type. Carotenoid levels ranged from an average of 25 to 30 mg/100 g FW (Figure 3e). These concentrations were in the ranges reported in [14,52,53] (17 to 40 mg/100 g FW) for spinach grown in soil.

### 3.5. Proline Accumulation

Leaf-blade proline (Pro) ranged on average from 3.22 to 4.27 mg/100 g FW (Figure 4). These values are lower than those recorded in [54] (4.66–43.15 mg/100 g FW) and are within the range recorded in [55] (2.74–7.2 mg/100 g FW) for spinach grown in the greenhouse and the open air. The proline concentration was higher in plants grown in coir pith than in those grown in the other substrates (Figure 4). Proline concentration is closely related to abiotic stress, such as water and nutrient deficiency and salinity [56–59]. This indicates that plants grown in coir pith may be subject to stress conditions. However, in the present study, proline did not correlate with growth parameters, such as leaf area and plant dry weight, as reported for young plants of tomatoes and lettuces in [60,61]. On the other hand, proline was negatively correlated to FRAP (r = −0.628, $p < 0.001$), which increases when the leaf extract's ability to reduce ferric iron decreases. The plant produces Pro to compensate for the oxidizing role of Fe, preventing the formation of reactive oxygen species (ROS). In chickpea, [62] also reported a negative correlation between proline content and FRAP.

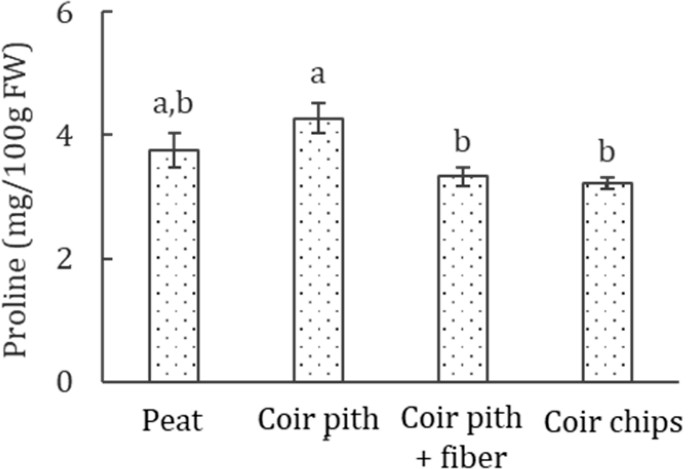

**Figure 4.** Proline content in the leaf blade. Means with different letters are significantly different at $p < 0.05$; FW—fresh weight. Each bar represents the mean of five replicates, and the error bars represent ± 1SE.

### 3.6. Phytochemical Accumulation

Leaf-blade TPCs, flavonoids, and GSH were higher in plants grown in coir pith than in those grown in the peat and the other coir types (Figure 5a,b,e).

Leaf-blade TPCs in plants grown in coir pith, peat, coir pith + fiber, and coir chips were 329, 263, 220, and 213 mg GAE/100 g FW, respectively (Figure 5a). TPC concentrations were next to the high end of the range reported by other authors (71–320 mg GAE/100 g FW) [6,63–65].

It is worth mentioning that total flavonoid content was significantly higher in plants grown in the different types of coir than in those grown in peat (Figure 5b). This could be due to the differences in leaf nutrient contents, shoot nutrient uptake, water availability of the substrate, irrigation scheduling, or interaction. Flavonoid biosynthesis is affected by nutrient and water availability and salinity [66,67]. The total flavonoids in the plants grown in different coir types ranged from an average of 6.58 to 7.14 mg/100 g FW. These values were higher than those recorded in [68] (1.45 to 4.47 mg/100 g FW) for 27 varieties of spinach grown organically and conventionally and were slightly lower than those recorded in [45] (8.25 mg/100 g FW). However, they were much lower than those reported in [69] (100 mg/ 100 g FW) and [70] (185 to 375 mg/100 g FW). The high variation might result from different genotypes investigated [68,71], maturation [70], etc.

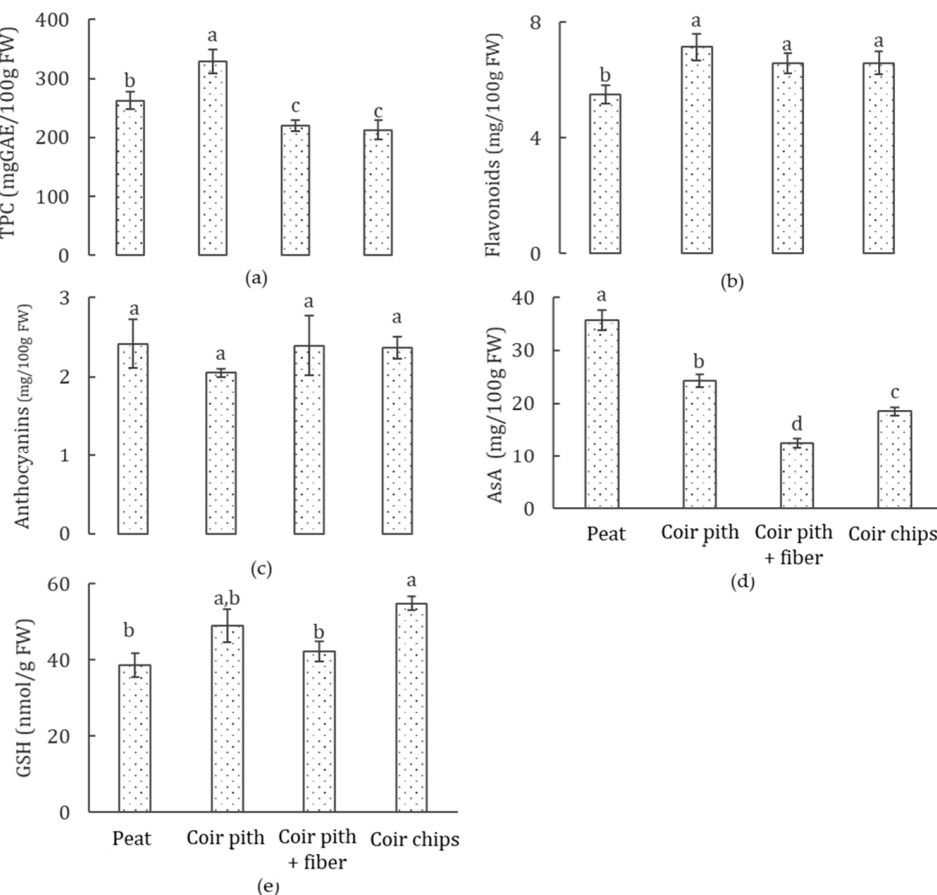

**Figure 5.** Contents of total phenolic compounds (TPCs) (**a**), flavonoids (**b**), anthocyanins (**c**), ascorbic acid (AsA) (**d**), and glutathione (GSH) (**e**) in the leaf blade. Means with different letters are significantly different at $p < 0.05$; FW—fresh weight. Each bar represents the mean of five replicates, and the error bars represent ± 1SE.

As a compound of the flavonoids group, the anthocyanin content ranged from an average of 2.05 to 2.42 mg/100 g FW. These values were similar to those recorded in [12] (15 to 38 mg/100 g DW), considering that the dry weight percentage of the spinach leaf-blade is close to 12%. However, the content was not significantly affected by the treatments (Figure 5c). This indicates that the higher antioxidant protection mediated by flavonoids in the plants grown in coir was affected by other flavonoid types.

AsA content of spinach grown in different growing media fell within the range reported by other authors (11 to 130 mg AsA/100 g FW) [14,68,72,73]. It was higher in leaf-blades of plants grown in peat (36 mg/100 g/FW) than in those grown in coir pith + fiber (13 mg/100 g/FW), coir pith (24 mg/100 g/FW), and coir chips (19 mg/100 g/FW) (Figure 5d). The differences could be related to leaf nitrogen content and/or plant nitrogen uptake since the nitrogen amount [74–76] and its form can affect AsA [53,77]. However, AsA was slightly correlated to leaf N (r = 0.483, $p < 0.05$). This may be due to the maintenance of ascorbic acid synthesis requiring a moderate amount of N [78].

Glutathione (GSH) content was higher in plants grown in coir than in those grown in peat. This indicates that plants grown in coir may have higher availability of antioxidant activity modulated by the SH group of cysteine [79]. ROS are scavenged by low-molecular-weight antioxidative metabolites like glutathione [80]. Leaf-blade GSH ranged from an average of 40 to 54 nmol/g FW (Figure 5e). These values are lower than those reported for spinach grown in soil in [81] (114–136 nmol/g FW), which could be related to the level of oxidative stress [79].

### 3.7. Antioxidant Enzyme (GR and POD) Activities

Leaf-blade GR activity was unaffected significantly by substrate type (Figure 6a). This means that the substrate type did not influence the glutathione tripeptide regeneration capacity of the spinach leaf blade. GR activity in the present study was approximately half of that reported in [82] for spinach (16.85 µmol min$^{-1}$/mg prot).

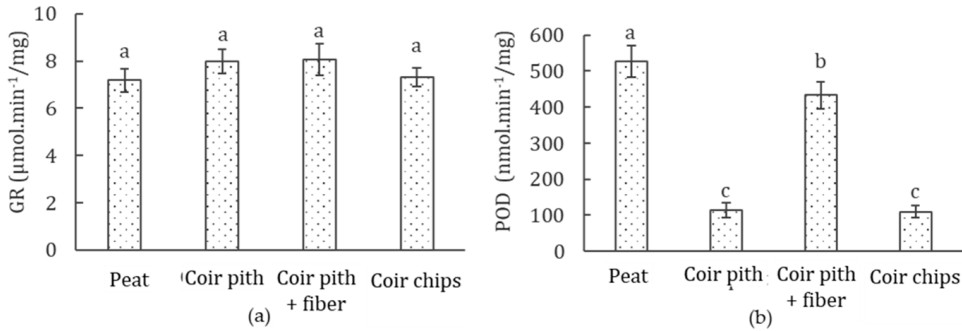

**Figure 6.** Glutathione reductase (GR) (**a**) activity and peroxidase (POD) (**b**) activity in the leaf blade. Means with different letters are significantly different at $p < 0.05$. Each bar represents the mean of five replicates, and the error bars represent $\pm$ 1SE.

POD activity was significantly higher in plants grown in peat than those grown in coir substrates (Figure 6b), reaching 527 nmol.min$^{-1}$/mg in peat, 433 nmol.min$^{-1}$/mg in coir pith + fiber, and 114 nmol.min$^{-1}$/mg in coir pith and coir chips. This could be related to the influence of substrates on plant nutrition and water uptake, as reported in [83]. The lower POD activity can be advantageous since peroxidases are the enzymes responsible for the browning of vegetable tissues [84]. Thus, spinach plants grown in coir substrate, mainly in coir pith and coir chips, may present a longer shelf life than those grown in peat. This is important in leafy vegetables since they are highly susceptible to enzymatic browning, shriveling, microbial growth, and loss of nutrients [85].

### 3.8. Antioxidant Activity

Leaf-blade FRAP was higher in plants grown in coir pith + fiber (32 mg Trolox/g FW) than in plants grown in peat (30 mg Trolox/g FW), coir chips (23 mg Trolox/g FW), and coir pith (10 Trolox/g FW) (Figure 7a). Generally, FRAP concentrations in our study were higher than those reported by other authors, which ranged from 2.67 to 13.8 Trolox/g FW [6,86,87]. The authors of [88] reported an increase in FRAP in basil as potassium increased in the nutrient solution. However, in the present study, FRAP was not correlated to leaf K. K concentration in nutrient solution affected total phenols, flavonoids, and antioxidant activity (FRAP, DPPH) in *Lavandula angustifolia* (Mill.). However, FRAP response to leaf K was not linear [89]. In the present study, despite K concentration in the nutrient solution being equal, leaf K was affected, but FRAP was not correlated to leaf K.

Leaf-blade DPPH was higher in plants grown in peat (38 mg GAE/100 g FW) than in plants grown in the coir types. Leaf-blade DPPH in plants grown in coir pith and coir chips ranged from 29 to 31 GAE/100 g FW (Figure 7b).

The free radical-scavenging activity estimated by DPPH has a strong positive correlation with AsA (r = 0.656, $p < 0.01$), indicating that ascorbic acid level contributes to the scavenging capacity of the leaf extract.

The differences in dry weight accumulation, phytochemical accumulation, antioxidant enzyme activities, and antioxidant power could be due to the characteristics of the substrates and/or effects of irrigation scheduling optimized to peat in water and nutrient uptake. Despite this, the findings indicate that the different types of coir, mainly coir pith, may provide a promising substitute for peat since it allowed reaching a high yield and increased total flavonoid content. The other phytonutrient contents and antioxidant activi-

ties were within the range of values reported in the literature for spinach. The adaptation of cultural management to the specific substrate and crop demand can further improve the quality of horticultural products [10,14,90]. Therefore, further research is needed to evaluate the response of spinach grown in different coir types with optimized irrigation and nutrient solution.

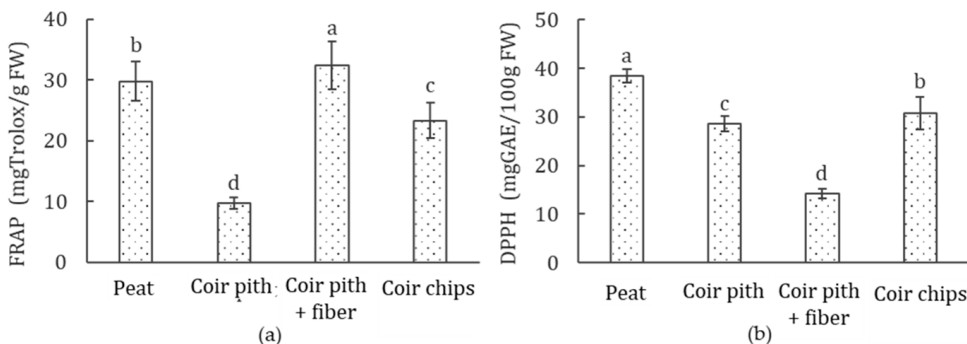

**Figure 7.** FRAP (**a**) and DPPH (**b**) in the leaf blade. Means with different letters are significantly different at $p < 0.05$; FW—fresh weight. Each bar represents the mean of five replicates, and the error bars represent ± 1SE.

## 4. Conclusions

Coir pith and coir pith + fiber may provide an alternative to peat. Plants grown in these substrates had a similar fresh yield but a higher total flavonoid content than plants cultivated in peat. The levels of other phytochemicals and the antioxidant activity (FRAP and DPPH) in plants grown in coir were within the usual ranges for spinach. However, further research will be necessary to analyze the effects of adjusting the irrigation scheduling and nutrient solution characteristics for each coir type, for instance, in coir chips, on spinach yield and product quality.

**Author Contributions:** R.M.A.M. conceived and designed the experiments; performed the experiments; analyzed and interpreted the data; contributed reagents, materials, analysis tools, or data; and wrote the paper. I.A.-P. and R.F. performed the experiments; analyzed and interpreted the data; contributed reagents, materials, analysis tools, or data; and wrote the paper. N.S.G. reviewed, corrected, and edited the paper. All authors have read and agreed to the published version of the manuscript.

**Funding:** This research was funded by Foundation for Science and Technology (FCT) as part of Project UIDB/05183/2020.

**Institutional Review Board Statement:** Not applicable.

**Informed Consent Statement:** Not applicable.

**Conflicts of Interest:** The authors declare no conflict of interest.

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
