# Peer review of "Coir, an Alternative to Peat—Effects on Plant Growth, Phytochemical Accumulation, and Antioxidant Power of Spinach"

_horticulturae, doi:10.3390/horticulturae7060127_

Round 1

Reviewer 1 Report

This study focuses on recycling coconut waste and coir in soilless vegetable crop production as a replacement for peat medium. The objective is obvious, and the result data clearly highlight the potential use of coir in vegetables to increase vegetable yield and enhance antioxidant activity. Still, it largely depends on the type of medium used. Overall, this study worth publication in the journal after careful revision on the following comments.

L82-87 – repeated sentences.

Fig. 2 legend – missed match between the legend and figure b c d, shoot dry weight (%) and shoot dry weight per plant. Five replicates instead of six

L264 – coir chip has the lowest leaf area and fresh yield compared to peat medium. Revise it.

L269-277. Authors describe less shoot biomass grown in coir medium than peat and discuss it due to less nutrient uptake and less water availability. It was understandable that less nutrient uptake might lead to reducing shoot biomass. However, it is hard to accept that the decrease in shoot biomass was partly due to less water availability in this study (L-116 - The nutrient solution was applied three to seven times per day) according to reference 38 because authors practice the same irrigation management for all treatments. There is no data to prove this claim. Revise discussion by focusing mainly on the growth medium like physicochemical properties of substrates such as CEC or other parameters.

L314-327 – If authors want to discuss high Zn concentration might be toxic to plants; it is better to show at which level of Zn in the tissue can occur Zn toxicity by citing other work.

L374-377 – authors discussed that lower chlorophyll content in plants grown in coir medium was related to high Zn concentration. However, it isn't easy to follow why authors discuss here suddenly salinity stress-related chlorophyll content. How do you connect salinity with your study? A more detailed explanation might need to follow your discussion.

L383-397 Increasing proline content in plants grown in coir medium without affecting fresh yield is one of the important results of this study. However, I can't entirely agree with your statement that ``This indicates that plants grown in coir pith may be subject to high-stress conditions``. If the plants experienced high-stress conditions, the fresh vegetable yield might dramatically reduce, but it was not the case in this study. Therefore, it would be interesting if authors revise this discussion part, and it should be mainly based on the interaction with Fe.

I have to read the method, again and again, to follow your abbreviations while working on results and discussions. Did the authors already mention the abbreviations of Pro and ROS in lines 395 and 396?

Revise axis titles in the figures (Fig. 2c, 3c, 3e, 4, 5c, 5e).

In conclusion

It is not correct for your conclusion that ``plants grown in coir had a higher yield and total flavonoid content than that in peat regardless of the type``. Please revise it. Not all types of coir positively respond to both fresh yield and phytonutrients. The authors need to answer here based on the objective and hypothesis set in the introduction part and specify which type of coir has the potential for replacement of non-renewable peat used in the vegetable crop production.

Author Response

L82-87 – repeated sentences. The repeated sentence was deleted

Fig. 2 legend – missed match between the legend and figure b c d, shoot dry weight (%) and shoot dry weight per plant. Five replicates instead of six

The legend was corrected

Six was replaced by five

L264 – coir chip has the lowest leaf area and fresh yield compared to peat medium. Revise it.

It was revised. Please find below the sentence revised

 “Leaf area and spinach fresh yield in coir pith and coir pith+fiber did not differ significantly from those obtained in peat (Figure 2). The coir pith and coir-pith + fiber yield were high ranging from 3.79 to 4.32 kg·m2.”

L269-277. Authors describe less shoot biomass grown in coir medium than peat and discuss it due to less nutrient uptake and less water availability. It was understandable that less nutrient uptake might lead to reducing shoot biomass. However, it is hard to accept that the decrease in shoot biomass was partly due to less water availability in this study (L-116 - The nutrient solution was applied three to seven times per day) according to reference 38 because authors practice the same irrigation management for all treatments. There is no data to prove this claim. Revise discussion by focusing mainly on the growth medium like physicochemical properties of substrates such as CEC or other parameters.

The possible influence of the physicochemical properties of substrates was inserted in the discussion as you can see in the following setenece: “Plants grown in coir accumulated less shoot biomass than those grown in peat. This can be due to the physiochemical properties of coir with less nutrient and water holding capacity as peat. Moreover, the initial content of nutrients in peat (Table 1) and the substrates' interaction “

L314-327 – If authors want to discuss high Zn concentration might be toxic to plants; it is better to show at which level of Zn in the tissue can occur Zn toxicity by citing other work.

We changed the sentence as you can see below:

” These Zn concentrations are higher than the sufficiency range (100 µg g-1 DM) [41] and (75 µg g-1 DM) [42]. However, none of the plants in the treatments showed visual symptoms of excess Zn. Zinc in excess can cause chlorosis in leaves due to a reduction in chlorophyll [43]. According to [44], leaf Zn concentrations of up to 100–700 mg kg−1 DM can be achieved without yield loss, which can be advantageous since Zn is a desirable nutrient for human health”

L374-377 – authors discussed that lower chlorophyll content in plants grown in coir medium was related to high Zn concentration. However, it isn't easy to follow why authors discuss here suddenly salinity stress-related chlorophyll content. How do you connect salinity with your study? A more detailed explanation might need to follow your discussion.

The sentence was changed to: “The differences in chlorophyll can be due to the water availability, salinity in the root media and nutrient uptake for nitrogen, potassium and zinc.”

L383-397 Increasing proline content in plants grown in coir medium without affecting fresh yield is one of the important results of this study. However, I can't entirely agree with your statement that ``This indicates that plants grown in coir pith may be subject to high-stress conditions``. If the plants experienced high-stress conditions, the fresh vegetable yield might dramatically reduce, but it was not the case in this study. Therefore, it would be interesting if authors revise this discussion part, and it should be mainly based on the interaction with Fe.

The sentence was changed to “This indicates that plants grown in coir pith may be subject to stress conditions. However, in the present study, proline did not correlate with growth parameters, such as leaf area, plant dry weight etc., as reported in young plants of tomatoes and lettuces by [60, 61].” However, as leaf proline was not related to leaf Fe, it is difficult to present a discussion based on the interaction with the iron.

I have to read the method, again and again, to follow your abbreviations while working on results and discussions. Did the authors already mention the abbreviations of Pro and ROS in lines 395 and 396?

Revised

Revise axis titles in the figures (Fig. 2c, 3c, 3e, 4, 5c, 5e).

Revised

In conclusion

It is not correct for your conclusion that ``plants grown in coir had a higher yield and total flavonoid content than that in peat regardless of the type``. Please revise it. Not all types of coir positively respond to both fresh yield and phytonutrients. The authors need to answer here based on the objective and hypothesis set in the introduction part and specify which type of coir has the potential for replacement of non-renewable peat used in the vegetable crop production.

The conclusion was revised to:

“The coir pith and coir pith + fiber may provide an alternative to peat. Plants grown coir pith and coir pith + fiber had a fresh yield similar to those obtained in peat and total flavonoid content higher than plant cultivated in peat. The levels of the other phytochemicals and the antioxidant activity (FRAP and DPPH) in plants grown in coir were within the usual range for spinach. However, further research will be necessary to analyze the effects of adjusting the irrigation scheduling and nutrient solution characteristics at each coir type (mainly in coir chips) on spinach yield and product quality.”

Reviewer 2 Report

Review of the manuscript: Coir an alternative to peat - effects on plant growth, phytochemical accumulation, and antioxidant power of spinach. 

In this study the  effects  of  four  commercial  substrates,a peat-based  substrate,and  three  coir  types (Coir pith, Coir chips,and Coirpith + fibers)on yield, phytochemical accumulation, and antioxidant activity were  evaluated  in Spinacia  oleraceaL.  cv. 'Manatee'. The article submitted for review is very important because the different types of coir may provide an alternative to peat. Plants grown in coir had a higher yield and total flavonoid content than plant cultivated in peat regardless of the type.

Introduction: The introduction is interesting, but in my opinion, it does not fully cover the topic. Below are some suggestions on how to expand this section. Moreover, out of 21 cited items, some are older than 10 years. Can these items not be replaced with newer ones?

Materials and Methods: Measurement of total phenol compounds (TPC): It must be pointed that only Folin-Ciocalteu method was used to identify the polyphenolic/phenolic content of the samples. It is a good way for comparing samples, but not to characterize them. More accurate methods could have been used. The same is the case: Chlorophylls and carotenoids: measurement and characterization by UV-VIS spectroscopy. Moreover, the authors refer to very old methods [24].

The topic is of interest and the experimental planning and execution are good. It should be noted that the authors described Materials and Methods very thoroughly, as well as the description and graphic representation of the results were presented very well. I believe that the work presented for review is of a high technical level. I am asking for a deeper description, taking into account my suggestions above, with post new items.

Author Response

Introduction: The introduction is interesting, but in my opinion, it does not fully cover the topic. Below are some suggestions on how to expand this section. Moreover, out of 21 cited items, some are older than 10 years. Can these items not be replaced with newer ones?

This introduction was improved by adding the following sentence “Social and ecological questions concerning child labor, inadequate wastewater management, and transportation should be additionally considered.”